# Effect on the Antioxidant Properties of Native Chilean Endemic Honeys Treated with Ionizing Radiation to Remove American Foulbrood Spores

**DOI:** 10.3390/foods13172710

**Published:** 2024-08-27

**Authors:** Enrique Mejías, Carlos Gomez, Tatiana Garrido

**Affiliations:** 1Centro de Tecnologías Nucleares en Ecosistemas Vulnerables, División de Investigación y Aplicaciones Nucleares—Comisión Chilena de Energía Nuclear, Nueva Bilbao 12501, Santiago 7600713, Chile; 2Departamento de Química Inorgánica y Analítica, Facultad de Ciencias Químicas y Farmacéuticas, Universidad de Chile, Olivos 1007, Independencia 8391063, Chile

**Keywords:** honey, botanical origin, *Paenibacillus larvae*, American foulbrood, phenolic compounds, antioxidants

## Abstract

In Chile, honey is produced from several native species with interesting biological properties. Accordingly, those attributes are present in Chilean honeys owing to the presence of phenolic compounds inherited from specific floral sources. In recent years, the exported volume of Chilean honeys has been increased, reaching new markets with demanding regulations directed toward the fulfilment of consumers’ expectations. Accordingly, there are countries with special requirements referring to *Paenibacillus larvae* spore-free honeys. This microorganism is the pathogen responsible for American foulbrood disease in beehives; however, antibiotics are not allowed when an apiary tests positive for *P. larvae*. On the other hand, it is mandatory to have an accurate method to remove the potential presence of spores in bee products intended for export. Exposure to ionizing radiation can be an efficient way to achieve this goal. In this work, 54 honey samples harvested from northern, central and southern Chile were analyzed for physicochemical patterns, total phenols, antioxidant activity and antiradical activity. Honeys with and without spores were exposed to ionizing radiation at three levels of intensity. Afterwards, the presence of spores and the effect on phenol bioavailability, antiradical activity and antioxidant activity were measured again. This research presents results showing a positive correlation between the percentage of prevalence of native endemic species in the set of honeys analyzed and the capacity to resist this process, without altering their natural attributes determined before irradiation treatments.

## 1. Introduction

Honey produced by bees, *Apis mellifera*, inherits the properties and biological attributes of its original floral source [1,2]. Accordingly, the nectar obtained by bees contains secondary metabolites of the honey’s plant of origin, mainly related to phenolic compounds and flavonoids, which are transferred to the final content of the honey and give it attributes such as antibiotic and/or antioxidant activity [3,4,5].

The production of honeys with potential natural value is an especially important fact for beekeepers, exporters and consumers [6] who are looking for natural, authentic products with high safety standards [7,8]

In Chile, beekeeping activity is distributed throughout the country [9]. At present, according to official records, there are more than 11,500 beekeepers who manage over one million five hundred thousand beehives [10], giving rise to a significant number of honeys and bee products with interesting biological properties due to the presence of a very varied endemic native flora. These include antibacterial and antioxidant properties and specific physicochemical parameters, among others [11]. In fact, the annual honey production in Chile is about 12,000 tons per year, and the exports of Chilean honey have reached up to 80% of this total amount [12]. Likewise, during the first half of 2024, economic growth of 26% compared to the same period of 2023 was observed for honey exports as stated by The Annual Report of the Office of Agricultural Studies and Policies of the Ministry of Agriculture of Chile [13]. Moreover, in comparison with 2023, there was an increase of 66% in volume of honey exported. In particular, 2776 tons of honey was shipped to Germany, the USA and France, the main destination countries [14].

However, beekeeping activity has faced various adverse conditions, including improper applications of agrochemicals [15,16,17,18] and the effects associated with climate change, whose impacts are mainly reflected in a lack of water and reduced availability of food for bees due to droughts [19], fires [20] and changes in the flowering calendar of melliferous species [21].

As a result, beehives have been weakened, causing an increase in bee diseases and considerable losses for the sector due to the decrease in the bee population and in the production volumes of bee products [22].

American foulbrood disease is a pathology caused by the Gram-positive, spore-forming bacterium *Paenibacillus larvae*. The endospores are only infective to larvae, especially in the first two days of the larval stage [23]. Although the disease does not manifest itself in adult bees, the brood dies, and the hives are not replenished [24]. After ingestion of food (honey or pollen) contaminated with spores, bee larvae die when the bacteria reach the hemolymph after passing through the gut wall [25]. Furthermore, the spores are highly resistant even at high temperatures and can persist for years in the surrounding environment of the apiaries [26]. When an outbreak of the disease is confirmed in a hive, the action protocol is euthanasia and subsequent incineration of the contaminated material, since treatment with antibiotics is not allowed [27].

The disease was detected for the first time in South America in Buenos Aires in 1989 [28], and the first report in Chile was recorded in the Atacama Region in 2001. Since 2017, the National Program for American Foulbrood Control has implemented a list of actions to avoid consecutive outbreaks of the disease in places with reported cases after 2001, such as the Regions of Atacama, Valparaíso, Metropolitana, Libertador Bernardo O’Higgins, Maule, Biobío and Los Lagos. It should be noted the most of cases ocurred in 2018 and 2019 with 44 and 66 outbreaks, respectively. Nevertheless, in the last three years, a decreasing trend in the prevalence and frequency of this disease has been observed [29].

On the other hand, as already mentioned, most Chilean honey is exported to other countries, and today, there are markets that demand products free of American foulbrood spores. Recently, Chile has established trade agreements with countries to expand sales to new markets for the export of Chilean honey. In this sense, the Protocol of Inspection, Quarantine and Sanitation established between Chile and the People’s Republic of China defines in articles 4 and 11 as a requirement for Chilean honey the absence of *P. larvae* spores within a radius of 50 km around the apiaries from which the honey is obtained, as well as the absence of outbreaks of the disease in the same area for two years prior to export [30].

In other matters, gamma radiation is electromagnetic radiation emitted by an unstable nucleus of an atom during radioactive decay; this process emits an analogous type of radiation, specifically a short wavelength electromagnetic radiation, which can be emitted by secondary products resulting from atomic fission or come from radioactive isotopes [31].

The application of ionizing radiation to food matrixes is an accepted technology that improves safety and extends shelf life by eliminating the presence of pathogenic microorganisms that affect the health of consumers [32]. Likewise, this technique enables extended food preservation, eliminating undesired insects and delaying germination and maturation processes, thus avoiding spoilage. Thus, the FDA approves the irradiation of a wide variety of foods such as beef and poultry, shrimp, fresh fruits and vegetables, seeds, eggs, spices and condiments [33].

In a similar way, ionizing radiation has also been applied to honey for the purpose of sterilization. This treatment allows the elimination of bacteria, viruses and parasites. However, due to the variability of the botanical composition of this product, it is necessary to establish adequate doses that do not affect the natural properties of the honey [34,35].

In this sense, the development and evaluation of an effective methodology for the control, prevention and eradication of American foulbrood spores in honeys by using gamma irradiation could be a useful tool for the strengthening of sustainable national apiculture. Therefore, the aims of this study were to develop an effective methodology for the control of American foulbrood without affecting the natural quality of Chilean honeys and to evaluate and compare the effect on antioxidant parameters of honeys of native Chilean endemic species treated with different levels of ionizing radiation.

## 2. Materials and Methods

### 2.1. Honey Samples

Fifty-four honey samples (labeled 1 through 54) were harvested from beehives located between the Coquimbo (29°54′ S) and Los Lagos (42°36′ S) Regions of Chile during 2021–2022 in austral spring–summer. All of the samples were collected in glass jars from the apiaries of origin. Afterward, the honey samples were transported to the laboratory for storage at 20 °C in the dark until analyses.

### 2.2. Identification of Botanical Origin by Melissopalynological Analysis

For the quantitative analyses, the method described by Loveaux et al. [36] with some modifications [37] was followed, and all the botanical pollen was counted. For the qualitative analyses, acetolyzed slides were prepared with 20 g of honey. Then, an aliquot was diluted with 20 mL of warm distilled water (40 °C). The solution was placed in an appropriate tube and centrifuged at 3500 rpm for 10 min. The supernatant was discarded, and the pollen residue was deposited at the bottom of the tube and resuspended in 100 mL of distilled water. An aliquot of this suspension (20 mL) was added to a slide, 10 mL of Calberla solution was added (a solution of either basic fuchsin or diamond) and the slide was gently dried. Next, 15 mL of melted glycerinated gelatin was added to the mixture. Finally, for each sample of honey, the pollen grain residues were identified using an optical microscope with total magnifications of 400× and 1000×.

### 2.3. Microbiological Detection of Spores of Paenibacillus larvae in Honey Samples

A qualitative microbiologic assay for detecting spores of *P. larvae* in honey was carried out. Honey samples were heated to 45–50 °C and shaken to distribute any spores, and then each honey sample was diluted (1/1) in 0.01 M phosphate-buffered saline (PBS) at pH 7.2 in 0.9% NaCl solution. Subsequently, the samples were transferred to a centrifuge tube and centrifuged at 3000 RPM for 30 min. The supernatant was discarded, leaving 3 mL per tube, which was then vortexed and mixed for 1 min to resuspend the pellet. Finally, 100–200 μL of the sediment–liquid mixture was poured into Mueller–Hinton broth, yeast extract, potassium phosphate, glucose and pyruvate (MYPGP) agar at 37 °C for 7–8 days [38].

### 2.4. Dosimetric Assays

The irradiation treatments were carried out in a Noratom 3500 irradiator Oslo, Norway (60 Co source, activity of 5000 Ci and a cylindrical irradiation volume of 2 L) located at the Irradiation Lab (CCHEN). Before performing the treatments, the irradiation volume of the Noratom 3500 irradiator was characterized using the alanine dosimetry system, formulated according to ASTM 51607 Standard [39].

### 2.5. Preparation of Honey Solutions for Colorimetric Assays

A mass of 10 g of honey was diluted with 50 mL of distilled water to obtain a homogeneous solution. The samples were stored at room temperature until the colorimetric analyses were performed. The pH range for all of the solutions was 5.5–6.5. The same procedure was followed for all the honey samples after irradiation treatments [40].

### 2.6. Total Phenolic Compound Content

The assay described by Buratti et al. [41] and Mejias et al. [42] was applied with minor modifications. Two hundred microliters of a honey solution was mixed with 50 mL of Folin–Ciocalteu reagent (Merck, Darmstadt, Germany), and then 150 mL of 20% Na_2_CO_3_ (Merck) was added. Finally, distilled water was added to final volume of 1.00 mL. The absorbance at 765 nm was determined after 30 min. Gallic acid (Sigma-Aldrich, St. Louis, MO, USA) was used as a standard to derive the calibration curve (0–150 mg/mL). The results define the phenolic content expressed as mg of gallic acid equivalents/kg of sample.

### 2.7. Determination of Antiradical Activity—DPPH Assay

The procedure described by Meda et al. [43] was followed to determine the antiradical activity. The 1,1-diphenyl-2-picrylhydrazyl radical (DPPH) assay allowed us to establish the antiradical properties of the chemical compounds in honey by the assessment of the oxidant activity of DPPH. Seven hundred fifty microliters of each honey solution was mixed with 1.5 mL of the DPPH (Merck) radical in methanol (0.02 mg DPPH/mL MeOH). The absorbance was determined after 15 min at 517 nm. A blank sample was prepared with methanol. Ascorbic acid (Calbio-Chem, Darmstadt, Germany) was used as a standard to derive the calibration curve (1–10 mg/mL). The values for antiradical activity were expressed as mg of ascorbic acid equivalents/g of sample.

### 2.8. Determination of Antioxidant Capacity in Honey—FRAP Assay

The ferric reducing/antioxidant power (FRAP) assay was performed according to Bertoncelj et al. [44]. FRAP reagent was prepared by mixing 2.5 mL of 2,4,6-tripyridyls- triazine (TPTZ) (Sigma-Aldrich) (10 mM TPTZ/40 mM of HCl) with 2.5 mL of 20 mM FeCl3 (Merck). Finally, 25.0 mL of 0.3 M acetate buffer at pH 3.6 was added to the mixture. The FRAP reagent was made each time the assay was performed. To measure the antioxidant capacity of the honey extracts, 0.2 mL of each sample (each honey solution) was mixed with 1.8 mL of the FRAP reagent. The absorbance was determined after 10 min at 593 nm. FeSO_4_·7H_2_O (Riedel de Haën, Seelze, Germany) was used as a standard to derive the calibration curve (50–1000 mM). The values were expressed as mM Fe^+2^ equivalents/g of sample.

### 2.9. Statistical Analysis

An exploratory data analysis based on statistical graphics was carried out for the purposes of choosing an appropriate analysis and checking assumptions.

One-way analysis of variance (ANOVA) was used to compare the effect of dose on the assessed parameters (phenolic compound content, DPPH and FRAP assays), with α = 0.05.

In this case, the ANOVA did not consider the botanical origin of the samples. A two-way ANOVA with interaction was performed to include this factor, with α = 0.05.

To evaluate the effect by type (botanical species) of honey, an ANCOVA was carried out between the covariate Dose and the component “Species”.

All of the data were processed using R software version 4.4.1—R Development Core Team 2024. Graphs were generated using the ggplot2 package.

## 3. Results

### 3.1. Geographical and Botanical Origin of Honey Samples

The geographical and botanical origins of the studied honey samples are indicated in Table 1, which also shows the percentages of the three most predominant botanical species found in the analyses. Additionally, the total pollen grain content used for this percentage is shown.

### 3.2. Detection of Spores of P. larvae in Honey Content

The presence of spores in honey samples is shown in Table 2. The results are presented with the geographical origin of each honey sample.

### 3.3. Dosimetric Assays and Selection of Radiation Dose Levels

Before carrying out the tests to define an adequate dose for the elimination of *P. larvae* spores, a dosimetric curve was derived to define the radiation exposure times necessary to reach the doses used in this study with the honey matrix. From this test, three dose levels were established: a low dose level, equivalent to 1 kGy (15 min); a medium dose level, equivalent to 5 kGy (79 min) and a high dose level, equivalent to 9 kGy (141 min). The intensity range was selected by considering the valid Chilean regulations for food irradiation [45].

Once the irradiation levels were defined, each honey that contained *P. larvae* spores was subjected to the three levels separately in order to establish the lowest effective dose for the eradication of the spores, as shown in Figure 1.

In the same way, the honeys that did not present spores in their content were also irradiated at the three levels to evaluate potential changes in phenolic compounds and antioxidant patterns after different doses of radiation (Figure 2).

### 3.4. Phenolic Compound Content in Honeys before and after Irradiation Treatments

All of the honey samples were irradiated at the three previously defined radiation levels. Figure 3 shows a comparison between the phenolic compound content in non-irradiated honeys and the results obtained after receiving the low and the medium doses. The results obtained after irradiation at the high dose are not included because the lowest effective dose was defined as the medium dose (5 kGy). Although the high dose was effective in controlling spores and there were no apparent changes in the samples irradiated with 9 kGy, there were acute changes in the phenolic compound content of some samples, A similar trend was observed for the other parameters measured. Thus, this level of irradiation was discarded for the other assays with the group of spore-free honeys. Neither geographical nor botanical origin is considered in this graph.

Regarding to the botanical origin, according to the Chilean regulation, thirty-nine of the total set of honeys analyzed in this study were found to be monofloral. Thus, Figure 4 shows the results concerning the variation of phenolic compound content in the honeys taking into consideration the monofloral botanical origin of the samples.

### 3.5. Antiradical and Antioxidant Activities in Honeys before and after Radiation Treatments

In a similar way, an analysis of antioxidants and antiradicals was conducted in the non-irradiated honeys, and the results were compared with the values obtained after the irradiation assays at low and medium levels. Figure 5 shows the results regarding antiradical activity in honeys without considering the botanical origin, and Figure 6 includes this factor as a differentiating element.

Finally, Figure 7 and Figure 8 show the results obtained for the antioxidant capacity (FRAP assay), presented schematically, as well as the antiradical activity.

## 4. Discussion

*Paenibacillus larvae* spores are highly resistant to the usual disinfection treatments and remain dormant for long periods of time. This makes it difficult to control the disease in hives when it is detected [46]. Therefore, the evaluation of alternative treatments that effectively eliminate the spores would help beekeepers to recover honey production in those apiaries that may show symptoms of the disease [47,48,49]. While it is true that adult bees carrying the spores cannot be irradiated, the removal of the spores from the hive products offers a real possibility of reducing the number of spores circulating in the environment around healthy hives [50].

In this study, it was possible to evaluate the effect of three irradiation levels on the honeys selected for analysis. It is important to mention that the radiation doses used are within the intensity range that has been frequently used in other food matrices [51].

At present, the valorization of honey as a natural food is mainly based on the natural attributes it inherits from the specific floral source [52]. The same cause is responsible for honeys having different consistencies, colors and flavors [53]. For this reason, it is not appropriate to consider the honey matrix a homogeneous matrix without considering these aspects when validating a methodology that seeks to eliminate American foulbrood spores contained in it [54].

In this sense, Chile has a diversity of landscapes and flora that vary from north to south [55]. Among the advantages of this biodiversity is the production of different honeys throughout the country [56]. For this reason, it was possible to select several honey samples from different productive regions with the assurance of having a wide variety of samples with very different native botanical compositions. Although multifloral honeys of introduced species are also produced in Chile, our goal was to obtain honeys from apiaries or beehives that had abundant native endemic forests in the surrounded areas. The main purpose of this part of the study was to select honeys that cannot be found in other parts of the world. This attribute of Chilean honey turns it into an attractive natural product to the international markets where it is exported. In this way, we obtained fifty-four honey samples from the most relevant regions in honey production in Chile. Thirty-nine samples tested as monofloral from native Chilean endemic species, while the rest of the honeys presented at least two native species as the main components of their content (Table 1).

The results obtained regarding the irradiations suggest that, despite this observed difference in botanical composition, the removal of *P. larvae* spores was optimal with doses of 5 kGy in those honeys that tested positive for its presence. Although the high dose used in this study (9 kGy) was also effective and no significant organoleptic differences were observed among the samples (Figure 2), 40% of the honeys irradiated with that dose showed an acute decrease in the quantity of phenolic compounds and changes in antioxidant capacity and antiradical activity without a clear trend in this response. The cause of this change may be a high rate of degradation of the secondary metabolites available in the content of the nectar from melliferous plants where bees take for producing these honeys. Those compounds are responsible for any given biological activity [57].

Although these compounds have a common metabolic pathway in plants, the different melliferous species described in this study are distributed in very different geographical locations that determine exposure to different climatic conditions; therefore, the honeys originating from these plants may react in contradictory ways to irradiation treatment at high doses. It should be noted that the irradiation processes did not raise the temperature of the honeys during the procedure, excluding this fact as an issue responsible for the results observed after the application of the radiation doses.

Foods that are frequently irradiated show resistance thresholds that, once reached, can generate an acute destruction of the original matrix. In the case of honey, the dose of 5 kGy would be within the threshold that allowed the elimination of spores without drastically modifying the original composition of the honey [58].

This makes sense and is reaffirmed by the results obtained for the content of phenolic compounds before and after irradiation in honeys (Figure 3). In general terms, when the results were only compared without incorporating the botanical origin in the data analysis, it was observed that there was a slight increasing trend in the amount of these compounds as the irradiation dose increased. The degradation of polyphenols due to exposure to ionizing radiation would explain this increase. However, when the data were analyzed considering the botanical origin of the samples (Figure 4), the tendency of the phenolic compounds to increase as a function of the increase in the radiation dose occurred in practically all the honey samples. Additionally, this effect was more intense in honeys obtained from the species *Luma apiculata* and *Prosopis chilensis*, which have high amounts of complex phenolic compounds with high antioxidant activity [59,60].

In turn, when the honeys were analyzed without considering botanical origin as a discriminating or differentiating element, no significant differences were observed between the honeys with and without irradiation treatment in comparison with the values obtained for antiradical activity and antioxidant capacity (Figure 5 and Figure 7). Nevertheless, the differentiation by botanical origin allowed us to establish a clearer trend in the behavior of the honeys in terms of antiradical capacity (DPPH assay) according to the radiation doses received (Figure 6). The results suggest the presence of two groups after the irradiation process. The first group is composed of *Luma apiculata*, *Brassica rapa*, *Eucryphia cordifolia*, *Caldcluvia paniculata* and *Tepualia stipularis*, which showed a minor decreasing trend in antiradical activity without showing significant differences between treatments. The second group is formed by the monofloral honeys of the species that showed no differences in antiradical activity before and after treatment with the radiation doses used in this work.

Likewise, *Quillaja saponaria* is known to exhibit naturally high antioxidant activities [42], which would indicate that honeys possessing these attributes would resist irradiation processes even better without drastically affecting their original properties (Figure 8).

For the analysis of antiradical and antioxidant activity, the DPPH and FRAP methods were selected as standard methods, since there are enough bibliographic references related to honey analyses. An electron release mechanism is the main chemical basis of the DPPH assay [61], unlike FRAP assay, where redox reactions take place. In this last case, antioxidant species are the main involved compounds, but other matrix components, such as reducing sugars, may act as interferents (>65%). In other words, carbohydrates may contribute to the modification of antioxidant power after assessment by the FRAP method [62]. Additionally, it is important to mention that the statistical analysis of correlations between FRAP and DPPH results showed R = 0.8, *p*-value = 8.2 × 10^−12^ (α = 0.05).

The phenolic composition of honey depends mainly on its floral origin; in fact, it can be used as a tool for classification and authentication, especially in the case of monofloral varieties. In addition, phenolic compounds are divided into phenolic acids and flavonoids. The first group contains Caffeic acid, p-Coumaric acid, Gallic acid, Vallinic acid, Syringic acid, Chlorogenic acid and 4(dimethyl amino) Benzoic acid, among others, while the second group includes Apigenin, Genistein, Pinocembrin, Chrysin, Quercetin, Luteolin, Kaempferol, Galangin and Pinobanksin [63]. The determination of phenols content by the Folin–Ciocalteu assay is a method based on a redox reaction with low selectivity; hence, there would be interferences in the quantification. Eventually, this could explain our results with no clear trends in the slopes when the botanical origin of monofloral honeys was included.

Finally, it is important to note that honeys cannot be referred as 100% monofloral samples. In this sense, botanical analyses allow the construction of profiles for each honey, considering all the botanical species available and correlated with the vegetal environment of the beehives. The entire profile of botanical species present in each honey is responsible for the natural biological activity that it may have, and therefore, the magnitude or intensity will be modulated by the interaction of the compounds coming from all of the melliferous plants responsible for the origin of that honey. However, at present, there is a change in the flowering calendars that affects the availability of nectar for the bees. This issue generates modifications in the annual production of apiaries, because there are severe droughts and or intense rains year after year due to climate change. This phenomenon has an important impact in the offer of floral supply for bees [64]. Moreover, bees must fly longer distances in search of nectar, weakening them and consequently generating an increase in the prevalence of diseases such as American foulbrood. In Chile, although the disease has been kept under surveillance and control, a displacement of outbreaks to areas that had not had the disease in previous years has been observed.

## 5. Conclusions

This study allowed us to analyze honey samples from an extensive range of Chilean territory. In this way, we were able to verify the presence of spores in honeys produced in areas with previous reports of American foulbrood, but it was not widely distributed throughout the country. In this sense, we can conclude that ionizing radiation was able to control the presence of American foulbrood spores (*Paenibacillus larvae*). The dose used in this study was in the range commonly applied to food.It was evidenced that the phenol content and the antiradical activity (DPPH assay) were increased with the application of both doses (low and medium) with respect to the untreated honey, but no clear trend in this modification between the phenol content and the botanical origin of the monofloral honey samples was observed.Our results suggest the strategic need to define territories according to botanical, geographical and ecological criteria for the characterization of honeys and, therefore, to increase their value in terms of bioactive compounds and absence of bee diseases.

## Figures and Tables

**Figure 1 foods-13-02710-f001:**
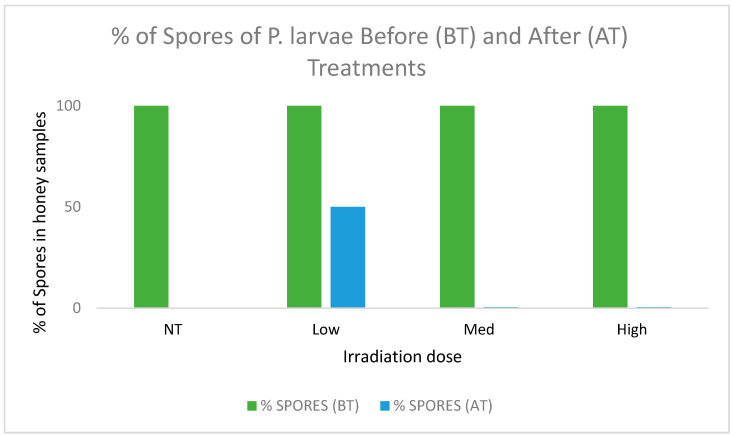
Percentage of spores of *P. larvae* in honey content before and after irradiation treatments. NT: no treatment; Low: 1 kGy; Med: 5 kGy; High: 9 kGy.

**Figure 2 foods-13-02710-f002:**
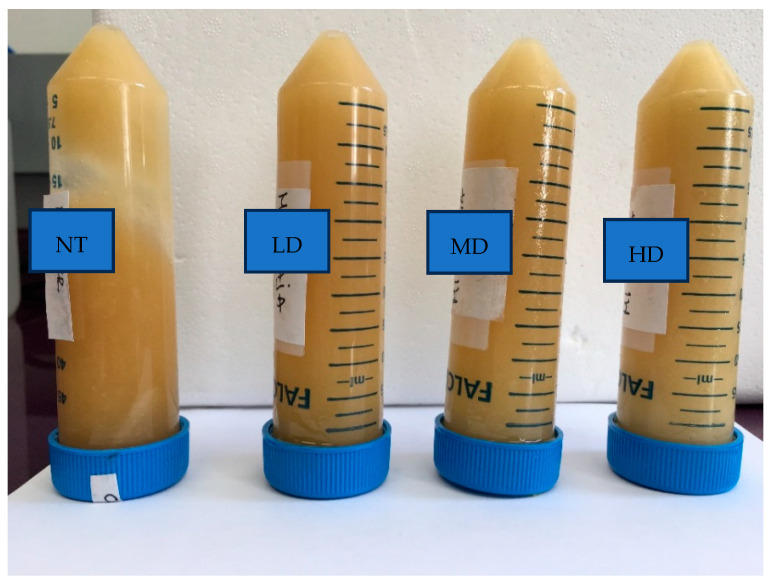
Evaluation of the effect on the organoleptic parameters of honey samples after irradiation assays at the three levels defined for this study. NT: no treatment; LD: low dose; MD: medium dose; HD: high dose.

**Figure 3 foods-13-02710-f003:**
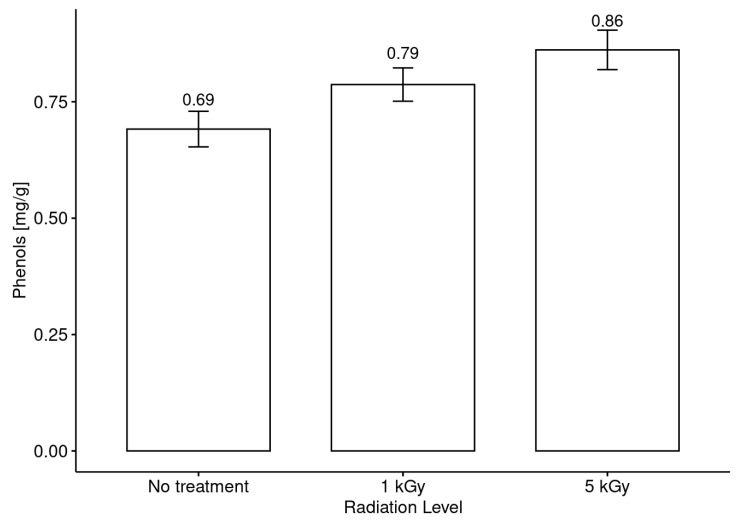
Changes in the phenolic compound content after irradiation treatments at low and medium doses.

**Figure 4 foods-13-02710-f004:**
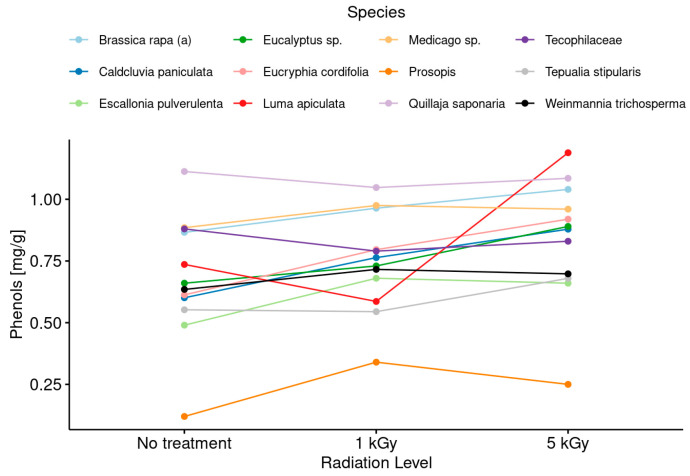
Changes in the phenolic compound content after irradiation treatments at low and medium doses considering the botanical origin of monofloral honey samples.

**Figure 5 foods-13-02710-f005:**
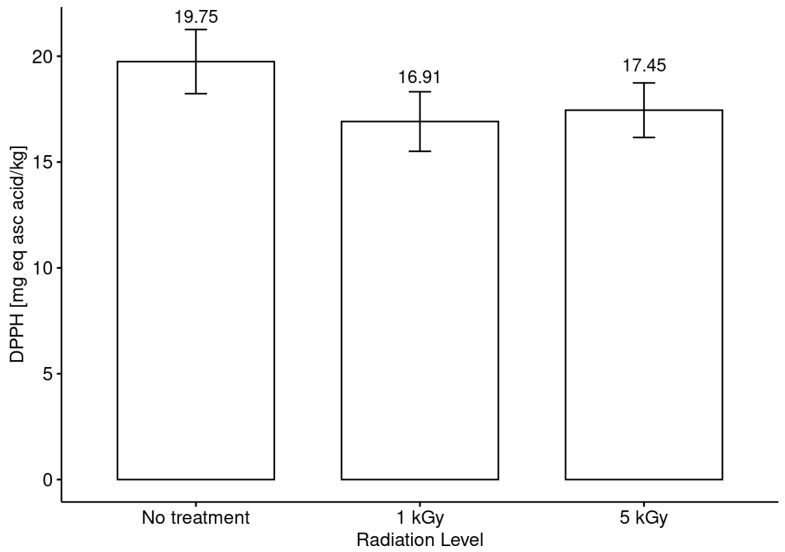
Antiradical activity (DPPH assay) assessment before and after irradiation treatment of honey at low and medium doses.

**Figure 6 foods-13-02710-f006:**
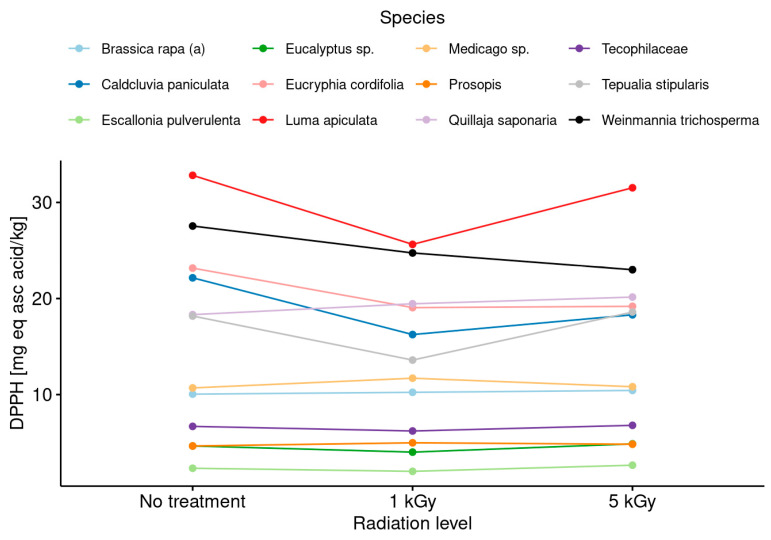
Changes in the antiradical pattern after irradiation treatments at low and medium doses considering the botanical origin of monofloral honey samples.

**Figure 7 foods-13-02710-f007:**
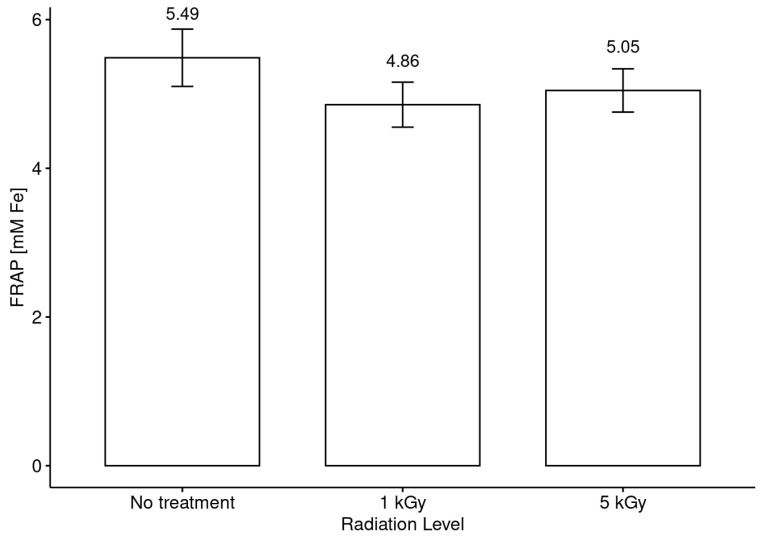
Antioxidant capacity (FRAP assay) assessment before and after irradiation treatment of honey at low and medium doses.

**Figure 8 foods-13-02710-f008:**
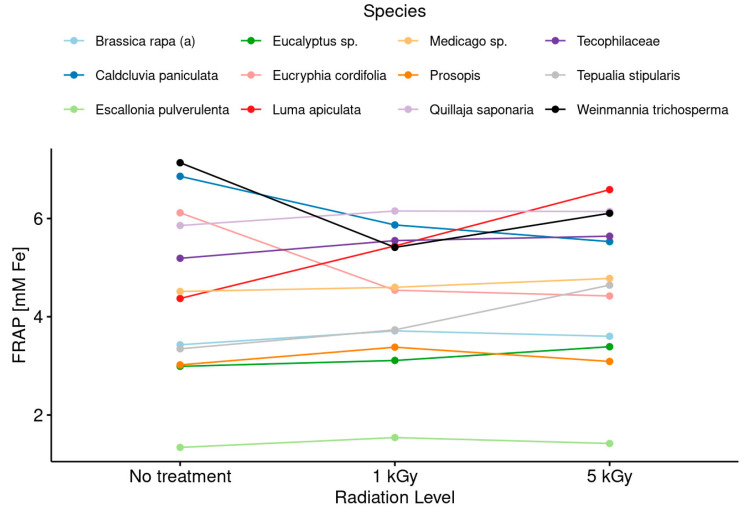
Changes in antioxidant capacity after irradiation treatments at low and medium doses considering the botanical origin of monofloral honey samples.

**Table 1 foods-13-02710-t001:** Geographical origin, predominant botanical species (%) and total grains of pollen content found in each honey sample.

Sample	Región of Chile	Species 1	Grains of Pollen (%)	Species 2	Grains of Pollen (%)	Species 3	Grains of Pollen (%)	Total Grains of Pollen	Presence of Spores of *P. larvae*
1	Coquimbo	*Escallonia pulverulenta*	67.49	*Olea/Citrus*	8.78	*Eucalyptus* sp.	6.29	683	Yes
2	Coquimbo	*Brassica rapa (a)*	33.22	*Eucalyptus* sp.	12.58	*Trifolium repens (a)*	8.71	620	Yes
3	Coquimbo	*Brassica rapa (a)*	74.13	*Eucalyptus* sp.	5.9	*Trifolium repens (a)*	4.54	661	Yes
4	Coquimbo	*Tecophilaceae*	33.39	*Quillaja saponaria*	28.87	*Lotus corniculatus*	7.9	620	Yes
5	Coquimbo	*Quillaja saponaria*	77.18	*Medicago* sp	6.25	*Trifolium repens (a)*	3.92	688	Yes
6	Coquimbo	*Prosopis*	69.09	*Brassica rapa*	10.86	*Tecophilaceae*	4.34	783	Yes
7	Metropolitana	*Medicago sp.*	15.95	*Luma chequen*	10.4	*Eucalyptus sp.*	9.11	702	No
8	Metropolitana	*Quillaja saponaria*	78.28	*Medicago* sp.	4.87	*Lotus corniculatus*	3.11	677	Yes
9	Metropolitana	*Brassica rapa (a)*	23.12	*Quillaja saponaria*	16.85	*Luma chequen*	13.65	713	No
10	Metropolitana	*Medicago* sp.	25.25	*Quillaja saponaria*	21.66	*Robinia pseudoacacia (a)*	11.47	697	No
11	Metropolitana	*Brassica rapa (a)*	22.73	*Tecophilaceae*	15.39	*Schinus polygamus*	11.36	695	No
12	Metropolitana	*Quillaja saponaria*	57.58	*Medicago sp.*	10.67	*Brassica rapa (a)*	4.21	712	No
13	Metropolitana	*Eucalyptus* sp.	21.25	*Robinia pseudoacacia (a)*	20.43	*Luma chequen*	18.39	734	No
14	Metropolitana	*Quillaja saponaria*	47.89	*Luma chequen*	19.34	*Medicago* sp.	12.45	697	No
15	Metropolitana	*Brassica rapa (a)*	39.67	*Quillaja saponaria*	15.34	*Schinus polygamus*	9.67	654	No
16	O´Higgins	*Galega officinalis (a)*	52.64	*Lotus corniculatus*	20.16	*Aristotelia chilensis*	5.12	625	No
17	O´Higgins	*Medicago* sp. *(a)*	31.58	*Lotus corniculatus*	25.27	*Galega officinalis (a)*	1001	649	No
18	Biobio	*Medicago* sp. *(a)*	38.76	*Lotus corniculatus*	12.16	*Aristotelia chilensis*	10.32	707	No
19	Biobio	*Castanea sativa*	55.52	*Caldcluvia paniculata*	12.42	*Medicago* sp.	7.52	652	No
20	Araucania	*Cladcluvia paniculata*	28.71	*Lotus pedunculatus*	23.76	*Escallonia rubra*	19.93	627	No
21	Los Rios	*Caldcluvia paniculata*	38.89	*Lotus pedunculatus*	26.58	*Buddleja globosa*	10.85	617	No
22	Los Rios	*Eucryphia cordifolia*	96.06	*Lotus pedunculatus*	1.57	*Buddleja globosa*	0.94	636	No
23	Los Rios	*Eucryphia cordifolia*	96.91	*Gevuina avellana*	1.21	*Luma apiculata*	1.07	744	No
24	Los Rios	*Caldcluvia paniculata*	77.11	*Lotus pedunculatus*	6.96	*Luma apiculata*	6.69	747	No
25	Los Rios	*Aristotelia chilensis*	49.84	*Lotus pedunculatus*	9.96	*Weinmannia trichosperma*	7.52	612	No
26	Los Rios	*Eucryphia cordifolia*	53.32	*Weinmannia trichosperma*	9.82	*Buddleja globosa*	8.61	662	No
27	Los Rios	*Caldcluvia paniculata*	45.43	*Buddleja globosa*	11.12	*Weinmannia trichosperma*	10.21	656	No
28	Los Rios	*Eucryphia cordifolia*	78.03	*Escallonia rubra*	3.23	*Weinmannia trichosperma*	3.07	651	No
29	Los Rios	*Weinmannia trichosperma*	87.26	*Caldcluvia paniculata*	9.22	*Buddleja globosa*	2.19	683	No
30	Los Rios	*Weinmannia trichosperma*	59.89	*Caldcluvia paniculata*	30.57	*Trifolium pratense (a)*	2.49	723	No
31	Los Rios	*Weinmannia trichosperma*	89.31	*Caldcluvia paniculata*	6.72	*Aristotelia chilensis*	2.14	655	No
32	Los Rios	*Weinmannia trichosperma*	88.13	*Aristotelia chilensis*	*10.71*	*Lotus pedunculatus*	0.43	691	No
33	Los Rios	*Weinmannia trichosperma*	95.65	*Aristotelia chilensis*	*2.39*	*Cissus striata*	0.9	667	No
34	Los Rios	*Weinmannia trichosperma*	55.83	*Caldcluvia paniculata*	*33.83*	*Buddleja globosa*	5.42	609	No
35	Los Rios	*Weinmannia trichosperma*	77.98	*Caldcluvia paniculata*	*12.66*	*Buddleja globosa*	3.58	727	No
36	Los Rios	*Caldcluvia paniculata*	77.46	*Azara microphylla*	*10.67*	*Lotus pedunculatus*	5.11	684	No
37	Los Rios	*Eucryphia cordifolia*	82.66	*Azara microphylla*	*14.22*	*Weinmannia trichosperma*	0.94	640	No
38	Los Rios	*Eucryphia cordifolia*	60.32	*Azara microphylla*	*32.67*	*Lotus pedunculatus*	3.73	698	No
39	Los Rios	*Weinmannia trichosperma*	86.34	*Caldcluvia paniculata*	*6.19*	*Aristotelia chilensis*	3.78	639	No
40	Los Lagos	*Tepualia stipularis*	52.42	*Lotus pedunculatus*	*17.79*	*Weinmannia trichosperma*	12.48	641	No
41	Los Lagos	*Tepualia stipularis*	58.23	*Lotus pedunculatus*	*23.61*	*Caldcluvia paniculata*	7.58	699	No
42	Los Lagos	*Trifolium repens*	43.08	*Medicago* sp.	*14.41*	*Lomatia* sp.	12.69	701	No
43	Los Lagos	*Caldcluvia paniculata*	90.32	*Lotus pedunculatus*	*3.99*	*Tepualia stipularis*	3.84	651	No
44	Los Lagos	*Luma apiculata*	59.38	*Trifolium repens*	18.32	*Tepualia stipularis*	5.91	677	No
45	Los Lagos	*Eucryphia cordifolia*	91.61	*Tepualia stipularis*	4.28	*Lotus pedunculatus*	3.67	655	No
46	Los Lagos	*Caldcluvia paniculata*	94.01	*Tepualia stipularis*	1.89	*Luma apiculata*	1.61	751	No
47	Los Lagos	*Eucryphia cordifolia*	89.74	*Tepualia stipularis*	4.24	*Weinmannia trichosperma*	2.61	614	No
48	Los Lagos	*Eucryphia cordifolia*	96.74	*Weinmannia trichosperma*	1.71	*Tepualia stipularis*	0.93	645	No
49	Los Lagos	*Eucryphia cordifola*	96.05	*Lotus pedunculatus*	1.11	*Trifolium pratense*	0.79	633	No
50	Los Lagos	*Tepualia stipularis*	31.53	*Luma apiculata*	26.72	*Caldcluvia paniculata*	24.82	685	No
51	Los Lagos	*Eucryphia cordifolia*	35.48	*Tepualia stipularis*	34.98	*Lotus pedunculatus*	11.39	606	No
52	Los Lagos	*Caldcluvia paniculata*	81.54	*Lotus pedunculatus*	6.27	*Tepualia stipularis*	5.96	665	No
53	Los Lagos	*Eucryphia cordifolia*	92.36	*Tepualia stipularis*	1.98	*Weinmannia trichosperma*	1.53	632	No
54	Los Lagos	*Eucryphia cordifolia*	96.74	*Weinmannia trichosperma*	1.71	*Tepualia stipularis*	0.93	681	No

a: Non native species.

**Table 2 foods-13-02710-t002:** List of samples positive for the presence of spores of *P. larvae* in the honey content.

Sample	Region of Chile	Presence of Spores of *P. larvae*
1	Coquimbo	Yes
2	Coquimbo	Yes
3	Coquimbo	Yes
4	Coquimbo	Yes
5	Coquimbo	Yes
6	Coquimbo	Yes
7	Metropolitana	No
8	Metropolitana	Yes
9	Metropolitana	No
10	Metropolitana	No
11	Metropolitana	No
12	Metropolitana	No
13	Metropolitana	No
14	Metropolitana	No
15	Metropolitana	No
16	O´Higgins	No
17	O´Higgins	No
18	Biobio	No
19	Biobio	No
20	Araucania	No
21	Los Rios	No
22	Los Rios	No
23	Los Rios	No
24	Los Rios	No
25	Los Rios	No
26	Los Rios	No
27	Los Rios	No
28	Los Rios	No
29	Los Rios	No
30	Los Rios	No
31	Los Rios	No
32	Los Rios	No
33	Los Rios	No
34	Los Rios	No
35	Los Rios	No
36	Los Rios	No
37	Los Rios	No
38	Los Rios	No
39	Los Rios	No
40	Los Lagos	No
41	Los Lagos	No
42	Los Lagos	No
43	Los Lagos	No
44	Los Lagos	No
45	Los Lagos	No
46	Los Lagos	No
47	Los Lagos	No
48	Los Lagos	No
49	Los Lagos	No
50	Los Lagos	No
51	Los Lagos	No
52	Los Lagos	No
53	Los Lagos	No
54	Los Lagos	No

## Data Availability

The original contributions presented in the study are included in the article, further inquiries can be directed to the corresponding author.

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
