# Peer review of "Effect on the Antioxidant Properties of Native Chilean Endemic Honeys Treated with Ionizing Radiation to Remove American Foulbrood Spores"

_foods, 2024, doi:10.3390/foods13172710_

Round 1
Reviewer 1 Report
Comments and Suggestions for Authors
The manuscript presents a very interesting study. However, there are some points that need to be addressed before the publication. The introduction must be improved by including very important information that helps to justify the study.
1. The introduction lacks important information. In the abstract, it is mentioned that “In recent years, the exportation volume of Chilean honey has been increased” However, the introduction doesn’t show the current production, current export levels, and their increase.
2. In addition, it is mentioned that there are considerable losses for the sector, but no numbers are mentioned to support this fact. How much has the bee population declined? What are the main reasons for this decrease? Include all the reasons for this behavior, not only the mentioned American foulbrood disease. What is the percentage of bee decline caused by American foulbrood disease? It is clear that this is not the only reason, so please mention all the factors that can affect the bee population before mentioning the AFD.
3. It is also important to mention the conditions for the growth of the microorganism of interest. What type of bacteria is it? What are its morphological characteristics and growth requirements?
4. Line 63. “Also, about 80% of honey production in Chile is exported”. Express in the volume of current production.
5. Line 63-64. “Today, there are markets that demand products free of American foulbrood spores”. Please specify what these markets are. Could they increase? It is important to support the performance of the work.
6. At the end of the introduction, the authors mention that gamma irradiation could be a useful tool for solving this problem. However, the authors need to include the definition of gamma irradiation. In addition, authors must add information related to previous studies carried out in the field of research. What other products have been treated with gamma irradiation? What were the results?
7. Materials. How were the samples stored? What materials were used?
8. Since the results for Detection of Spores in the samples were yes/no, it is necessary to make explicit from the methodology the qualitative character of the analysis performed. Otherwise, it would be important to see if there were differences between the samples marked as yes for the presence of spores.
9. Line 197-202. How the irradiation levels for preliminary assays were defined?
10. Figure 1. Percentage of spores of P. larvae in honey content before and after irradiation treatments. The information in the figure is not clear. Correct the 0 axes and specify which is before and after.
11. Line 219-220. “The results obtained after irradiation at the high dose are not included due to the lowest effective dose was defined as the medium dose (5 kGy)”. If the lowest effective dose is 5, this implies that the next dose was also effective. Why were the results not included for phenolic content? Please clarify.
12. Figure 3. Changes in the Phenolic Compounds Content after irradiation treatments at Low and Medium Dose. The figure is also not the most appropriate to show the phenomenon. I would suggest using bar charts. Same comment for Fig 5.
13. The discussion needs to be improved by mentioning the mechanisms of each antiradical and antioxidant activity used. Why DPPH and FRAP did have different results when compared to botanical sources? Why were these used and not other analyses? Was there any correlation between the behavior of the phenolic compounds and the activities evaluated? It is also important to mention the main phenolic compounds that are present in the honey.
14. The conclusion may be improved to highlight the scientific conclusion also and not only repeat the obtained results.
Author Response
Thank you for your valuable and helpful feedback. We have included all your suggestions and comments. Here, you will find the answers for every point you mentioned.
1. The introduction lacks important information. In the abstract, it is mentioned that “In recent years, the exportation volume of Chilean honey has been increased” However, the introduction doesn’t show the current production, current export levels, and their increase.
R. Thank you for this comment. We added information about this topic with appropriate references.
2. In addition, it is mentioned that there are considerable losses for the sector, but no numbers are mentioned to support this fact. How much has the bee population declined? What are the main reasons for this decrease? Include all the reasons for this behavior, not only the mentioned American foulbrood disease. What is the percentage of bee decline caused by American foulbrood disease? It is clear that this is not the only reason, so please mention all the factors that can affect the bee population before mentioning the AFD.
R. We agree with this point. In the original manuscript the list of factors that have contributed to the decline of bees is mentioned, however we incorporate a brief review of the behavior of the American foulbrood in Chile since it was first detected.
3. It is also important to mention the conditions for the growth of the microorganism of interest. What type of bacteria is it? What are its morphological characteristics and growth requirements?
R. Thank you for this suggestion. We improved the Introduction with additional background information.
4. Line 63. “Also, about 80% of honey production in Chile is exported”. Express in the volume of current production.
R. Thank you for this commented. We corrected and improved this data.
5. Line 63-64. “Today, there are markets that demand products free of American foulbrood spores”. Please specify what these markets are. Could they increase? It is important to support the performance of the work.
R. We added additional background for supporting those sentences.
6. At the end of the introduction, the authors mention that gamma irradiation could be a useful tool for solving this problem. However, the authors need to include the definition of gamma irradiation. In addition, authors must add information related to previous studies carried out in the field of research. What other products have been treated with gamma irradiation? What were the results?
R. Thank you for this suggestion. We improved the Introduction with additional background information.
7. Materials. How were the samples stored? What materials were used?
R. We have included more details in the description in the Methods section to clarify this point.
8. Since the results for Detection of Spores in the samples were yes/no, it is necessary to make explicit from the methodology the qualitative character of the analysis performed. Otherwise, it would be important to see if there were differences between the samples marked as yes for the presence of spores.
R. We agree with this comment, and we have included your suggestion in the Methods section.
9. Line 197-202. How the irradiation levels for preliminary assays were defined?
R. We added additional background for supporting this regarding to Chilean Regulations for Food Irradiation. This is available in the new version of our manuscript.
10. Figure 1. Percentage of spores of P. larvae in honey content before and after irradiation treatments. The information in the figure is not clear. Correct the 0 axes and specify which is before and after.
R. Thank you for this comment. We changed this figure.
11. Line 219-220. “The results obtained after irradiation at the high dose are not included due to the lowest effective dose was defined as the medium dose (5 kGy)”. If the lowest effective dose is 5, this implies that the next dose was also effective. Why were the results not included for phenolic content? Please clarify.
R. We added additional explanation to clarify this misunderstanding.
12. Figure 3. Changes in the Phenolic Compounds Content after irradiation treatments at Low and Medium Dose. The figure is also not the most appropriate to show the phenomenon. I would suggest using bar charts. Same comment for Fig 5.
R. Thank you for these suggestions. We changed the figures.
13. The discussion needs to be improved by mentioning the mechanisms of each antiradical and antioxidant activity used. Why DPPH and FRAP did have different results when compared to botanical sources? Why were these used and not other analyses? Was there any correlation between the behavior of the phenolic compounds and the activities evaluated? It is also important to mention the main phenolic compounds that are present in the honey.
R. We improved the discussion, added more supporting information and extended the comments. We hope these improvements will be satisfactory to you.
14. The conclusion may be improved to highlight the scientific conclusion also and not only repeat the obtained results.
R. A similar response. We introduced changes for improving the soundness of conclusions.

Reviewer 2 Report
Comments and Suggestions for Authors
Report on “Effect on the antioxidant properties of native Chilean endemic honeys treated with ionizing radiation to remove American foulbrood spores” by Mejías et al. In foods as a research article.
In this article, the authors describe in the gamma irradiation of different honey varieties from different country regions in Chile with regard to the spore destruction in the honey and the honey quality based on total phenolic content, antiradical and -oxidant capacity.
The safety and quality of honey, especially if exported, is important and sometimes required. Therefore, this work gives an insight in the possibility to treat the honey with gamma irradiation to guarantee the food safety without harming the quality deeply.
However, the authors should differentiate more between the two challenges A) bacterial infection of the bees due to environment or external intake with the result of loss of bee populations and B) the contaminated honey which cannot be exported.
In general, the application of gamma irradiation may work for the honey, but not for the bees.
Further, the authors may include some words for the legal and economic side of gamma irradiation on honey. Is the treatment allowed worldwide or some regions and therefore maybe the export possibilities limited?
The authors may also provide toxicological studies of gamma irradiated honey from literature, if available, to show consumers safety.
From the economical point of view, will the treatment increase the market prices and will this be accepted from the market?
Concerning the graphs:
It seems that the results are combined for all investigated honeys. Why? Can you please give more explanation?
Are the data for each single honey available in the supplementary later?
The article is good to read and meets the formal requirements. However, the English could be improved to be in some cases clearer.
Due to the above given comments, a major revision is recommended.
Comments on the Quality of English Languageplease see comments above
Author Response
Dear Reviewer,
Thank you for your valuable and helpful feedback. We have included all your suggestions and comments. Here, you will find the answers for every point you mentioned:
In this article, the authors describe in the gamma irradiation of different honey varieties from different country regions in Chile with regard to the spore destruction in the honey and the honey quality based on total phenolic content, antiradical and -oxidant capacity.
The safety and quality of honey, especially if exported, is important and sometimes required. Therefore, this work gives an insight in the possibility to treat the honey with gamma irradiation to guarantee the food safety without harming the quality deeply.
However, the authors should differentiate more between the two challenges A) bacterial infection of the bees due to environment or external intake with the result of loss of bee populations and B) the contaminated honey which cannot be exported.
R. We enriched the introduction section with additional background information with appropriate references. We hope these improvements will be satisfactory to you.
In general, the application of gamma irradiation may work for the honey, but not for the bees.
R. We introduced additional information for gamma irradiation, food irradiation and Chilean regulations.
Further, the authors may include some words for the legal and economic side of gamma irradiation on honey. Is the treatment allowed worldwide or some regions and therefore maybe the export possibilities limited?
R. This was included. We have included additional information on FDA's recommendations for irradiated foods.
The authors may also provide toxicological studies of gamma irradiated honey from literature, if available, to show consumers safety.
R. We added in both sections (introduction and discussion) additional information on irradiated honeys and effects.
From the economical point of view, will the treatment increase the market prices and will this be accepted from the market?
R. Thank you for this interesting comment. By the way, any type of treatment involves an increase in cost, but also brings benefits from the point of view of availability, longer storage time and improved hygiene. For the moment, no cost estimates are available since irradiations in Chile can be performed in state facilities and could be subsidized if required. Our purpose is to offer an alternative in the treatment of honeys eventually contaminated with American foulbrood if needed.
Concerning the graphs:
It seems that the results are combined for all investigated honeys. Why? Can you please give more explanation?
R. We changed some figures, and we clarified the statistical analyses. Also, we enriched the Discussion with additional background.
Are the data for each single honey available in the supplementary later?
R. Thank you for this comment. We could include them, although we believe that with the improvements incorporated in the figures and discussion, additional numerical data would not be necessary and would not provide more background.
The article is good to read and meets the formal requirements. However, the English could be improved to be in some cases clearer.
R. We reviewed the English grammar with a native speaker who made some corrections that we hope will be to your satisfaction.

Round 2
Reviewer 1 Report
Comments and Suggestions for Authors
The authors have improved the manuscript according to the suggestions. Therefore, I accept the manuscript in its current form.
Author Response
- The authors have improved the manuscript according to the suggestions. Therefore, I accept the manuscript in its current form.
Thank you for this kind revision.
Reviewer 2 Report
Comments and Suggestions for Authors
no further comments
Author Response
- No further comments.
Thank you for this kind revision